# Polymeric Ionic Liquids Derived from *L*-Valine for the Preparation of Highly Selective Silica-Supported Stationary Phases in Gas Chromatography

**DOI:** 10.3390/polym12102348

**Published:** 2020-10-14

**Authors:** Jorge González-Rodríguez, Adriana Valls, Pilar Arias Abrodo, María Dolores Gutiérrez Álvarez, Jaime González-Álvarez, Belén Altava, Santiago V. Luis

**Affiliations:** 1Department of Physical and Analytical Chemistry, University of Oviedo, Julián Clavería 8, 33006 Oviedo, Spain; jorgeglez7@gmail.com (J.G.-R.); piarab@uniovi.es (P.A.A.); loly@uniovi.es (M.D.G.Á.); 2Department of Organic and Inorganic Chemistry, University Jaume I, Avda. V. Sos Baynat, 12071 Castellón, Spain; avalls@uji.es

**Keywords:** gas chromatography, ionic liquids, cross-linked stationary phases, solvation parameter model

## Abstract

A series of silica-supported polymeric ionic liquid (PIL)-based stationary phases derived from a vinylic *L*-valine ionic liquid monomer and divinylbenzene (DVB) as the crosslinking agent have been prepared and studied as gas chromatographic stationary phases. These coated gas chromatographic columns exhibited good thermal stabilities (230–300 °C) and high efficiencies (1700–2700 plates/m), and were characterized using a linear solvation parameter model in order to understand the effects of the amount of DVB on the features of the resulting composite systems. Their retention behavior and separation efficiencies were demonstrated using the Grob test. By tuning the crosslinking degree for the IL-derived stationary phase, the separation selectivity and resolution of different compounds were improved. The different retention behaviors observed for many analytes indicate that these stationary phases may be applicable as new types of GC stationary phases.

## 1. Introduction

Ionic liquids (ILs) present unique features that, along with their structural modularity, favor their efficient use for a variety of applications and chemical processes [1,2,3]. However, toxicological or environmental concerns [4], as well as some of their physico-chemical properties (like high viscosities) or their high cost, have often hampered their practical application. Many of those limitations can be overcome with the use of materials based on ILs [5]. In this context, polymeric ionic liquids (PILs) have been shown to present remarkable properties and applications [6,7,8,9,10,11,12], enabling new and interesting separation processes [13,14,15].

In the search of improved properties like negligible vapor pressure and better-symmetry peaks, new types of stationary phases in gas chromatography (GC) are constantly being explored. In this context, ionic liquids (ILs) have drawn much attention in recent years as materials for stationary phases in GC, due to properties like their capacity to establish simultaneous polar and nonpolar interactions with the analytes, their high thermal stability, or their negligible vapor pressures and wide liquid ranges [16,17,18,19,20]. Besides, it is worth mentioning that these properties can be easily fine-tuned through small changes in the structure of both the cation or the anion, which, besides, can dramatically change the selectivity or the separation capacity for the analyte of interest [21,22,23,24].

Thus, for instance, the increase in the nucleophilicity of the anion can decrease the thermal stability of the corresponding ionic liquids, while its coordinating capacity can define the quality of the chromatographic separation. In this regard, stationary phases based on ionic liquids with anion halides tend to strongly retain analytes displaying hydrogen bonding donor abilities (for example, alcohols and acids), resulting in long retention times and peak asymmetry. Conversely, ILs-based stationary phases containing noncoordinating anions such as hexafluorophosphate in the IL structure often result in poor separations and overlapping peaks. In the case of ILs containing the bis[(trifluoromethyl)sulfonyl]imide (NTf_2_^−^) anion, the resulting stationary phases have been shown to exhibit higher thermal stability and lower surface tension as compared to ILs encompassing other anions such as tetrafluoroborate or hexafluorophosphate [25,26,27].

This modular character of ILs allows for a tailor-made design of new structures intending to achieve specific properties (task-specific ILs-TSILS) [28]. This can be illustrated by amino acid-based ionic liquids (AAILs). The preparation of a variety of AAILs has been reported in the literature, which allows the introduction of additional functionalities like carboxylic acid, amino, amide, or alcohol groups in the IL structure [29,30,31,32]. These AAILs have additional functional groups able to provide complementary interactions such as hydrogen bonding or hydrophobic and aromatic interactions that make them interesting molecules in supramolecular chemistry being used as selective receptors or transport agents through pseudobiological membranes [33,34]. These features also provide a strong potential for their use as stationary phases in GC, leading to the recent preparation and characterization of some new AAILs-based gas chromatography stationary phases, with good potential for the separation of the fatty acid derivatives [35].

The interest in IL-coated GC columns has increased after their commercial introduction in 2008, and nowadays, several IL-coated columns with different characteristics are commercially available. However, one major challenge for the development of stationary phases based on ILs is the preparation of highly homogenous coatings, which would favor good peak symmetries and highly effective compound separations, and concomitantly, provide high thermal stabilities for the resulting GC columns [36,37,38]. At high temperatures, uniform IL-coated silica columns can experience film disruption leading to a decrease in the analyte retention times and efficiency. In this regard, polymerized ionic liquids (PILs) can provide the required answers, maintaining the excellent thermal stability of the columns, and combining the main features of an ionic liquid and the typical polymer properties such as improved mechanical stability and processability [39,40,41,42]. Furthermore, in recent years, Anderson et al. reported the development of cross-linked polymeric ionic liquid stationary phases for GC with higher thermal stability (up to 380 °C) [43] than the columns coated with the monocationic or dicationic ILs with similar structures [44]. More recently, this research group developed different crosslinked PILs in order to better understand the effects of the crosslinker on the efficiency, selectivity, and thermal stability of PILs-based stationary phases [45]. These studies have focused on the use of dicationic ILs as the crosslinker agent. However, the use of a simple, cheap, and commercially available crosslinker like divinylbenzene (DVB) can provide significant advantages, as the use of functional crosslinkers can be of lower efficiency, considering the reduced accessibility associated with functional moieties in the crosslinked regions. Thus, the use of the DVB crosslinker can be used as a simple design element to tune the morphological properties of the resulting polymeric matrices. As far as we know, the use of crosslinked PILs-based stationary phases for GC using DVB as a crosslinker agent has not been studied.

In this work, a series of silica-supported polymeric ionic liquid (PIL)-based stationary phases have been prepared using a vinylic *L*-valine ionic liquid derivative (AAIL) as a monomer and divinylbenzene (DVB) as a crosslinking agent in order to provide an homogeneous coating of the silica surface and improve the column efficiency. The thermal stabilities of these stationary phases based on AAIL-DVB were evaluated, and the Abraham model was used in order to determine their solvation parameters. Furthermore, the effect of the amount of the AAIL and the crosslinker was analyzed and carefully optimized to improve the retention, selectivity, and resolution of the columns using the Grob test.

## 2. Materials and Methods

### 2.1. Reagents, Materials, and Instrumentation

Divinylbenzene and all reagents used for the synthesis of the monomeric AAIL, as well as the Grob mixture, were purchased from Sigma-Aldrich (Madrid, Spain). Methanol and *n*-dichloromethane were obtained from Merck (Darmstadt, Germany), and untreated fused silica capillaries (0.25 mm i.d.) were purchased from Supelco (Madrid, Spain). Imidazole **2** was synthesized as previously described starting from the corresponding amino amide **1** (Figure 1) [46].

NMR spectra were recorded on a Varian INOVA 500 spectrometer (500 and 125 MHz for ^1^H and ^13^C NMR, respectively) (Agilent, Santa Clara, CA, USA) at 30 °C. The solvent signal was used as a reference standard and the chemical shifts are reported in ppm. FT-IR spectra were obtained at 4 cm^−1^ resolution for the 4000 to 600 cm^−1^ spectral range using a spectrometer (JASCO FT/IR-6200) equipped with an ATR (MIRacle single-reflection ATR diamond/ZnSe) (JASCO, Easton, MD, USA). Melting points were measured using a differential scanning calorimeter (DSC) (DSC6, Perkin Elmer). The instrument was calibrated for temperature and heat flow with zinc and indium reference samples. Samples were placed in a hermetically sealed aluminum pan with a pinhole in the top. An empty aluminum pan was used as the reference. Samples were exposed to a flowing N_2_ atmosphere. Before the DSC test, samples were dried at 60 °C in a vacuum oven for 12 h. Melting transition temperatures (M.p.) were determined by multiple cycles (typically three) involving heating the sample from −70 to 125 °C followed by cooling from 125 to −70 °C, both at a rate of 5 °C min^−1^. The melting temperatures were determined at the onset of the transition. Mass spectra were recorded on a Q-TOF Premier mass spectrometer with an orthogonal Z-spray electrospray interface (Micromass, Manchester, UK) by the electrospray positive mode (ES+). Sample solutions were infused via a syringe pump directly connected to the ESI source at a flow rate of 10 mL min^−1^. Microanalyses were performed on an elemental analyzer equipped with an oxygen module (CHN EuroEA3000) (LECO, Geleen, NL, USA). Optical rotations were measured in a JASCO instrument DIP-1000. (JASCO, Easton, MD, USA).

### 2.2. Chemical Synthesis

#### 2.2.1. Synthesis of (S)-3-(but-3-en-1-yl)-1-(1-(butylamino)-3-methyl-1-oxobutan-2-yl)-1H-imidazol-3-ium bromide 3

Synthesis of (S)-3-(but-3-en-1-yl)-1-(1-(butylamino)-3-methyl-1-oxobutan-2-yl)-1H-imidazol-3-ium bromide 3 (Scheme 1).

Compound **2** (0.10 g, 4.43 × 10^−4^ mol, 1 equivalent), obtained as previously described [46], and 4-bromo-1-butene (0.93 mL, 4.43 × 10^−4^ mol, 10 equivalens) were dissolved in dry acetonitrile (1 mL) in a round-bottom flask (25 mL). The reaction mixture was stirred and heated at 60 °C for 24 h under a N_2_ atmosphere. The solvent was evaporated under reduced pressure and the resulting crude residue was washed with hexane:ether (2:1), affording compound **3** as a yellow oil (0.0775 g, 47.2%). M.p. 9 °C. [α]${}_{D}^{25}=$ +27.64 (c = 0.033, MeOH). IR (ATR): ν_max_ = 3413, 3227, 3068, 2962, 2933, 2872, 1673, 1550, 1463, 1159 cm^−1^. ^1^H NMR (CDCl_3_) δ = 9.84 (s, 1H), 8.55 (s, 1H), 7.77 (s, 1H), 7.15 (s, 1H), 5.80–5.62 (m, 2H), 5.17–4.94 (m, 2H), 4.33–4.17 (m, 2H), 3.31–3.20 (m, 1H), 3.15–3.03 (m, 1H), 2.70–2.54 (m, 2H), 2.46–2.34 (m, 1H), 1.58–1.41 (m, 2H), 1.38–1.22 (m, 2H), 1.03 (d, *J* = 6.6 Hz, 3H), 0.83 (t, *J* = 7.3 Hz, 3H), 0.73 (d, *J* = 6.7 Hz, 3H). ^13^C NMR (CDCl_3_): δ = 167.0, 136.1, 131.5, 121.6, 121.0, 120.3, 67.6, 49.5, 39.6, 34.2, 31.3, 31.0, 20.2, 18.7, 18.2, 13.6 ppm. MS (ESI+): 278.4 [M + H^+^, 100]. C_16_H_28_BrN_3_O·1.5 H_2_O: calcd. C 53.63, H 7.88, N 11.73; found C 53.16, H 8.04, N 11.17.

#### 2.2.2. Synthesis of (S)-3-(but-3-en-1-yl)-1-(1-(butylamino)-3-methyl-1-oxobutan-2-yl)-1H-imidazol-3-ium triflamide 4 

Synthesis of (S)-3-(but-3-en-1-yl)-1-(1-(butylamino)-3-methyl-1-oxobutan-2-yl)-1H-imidazol-3-ium triflamide 4 (Scheme 1).

Compound **3** (0.12 g, 2.76 × 10^−4^ mol, 1 equivalent) and lithium bis((trifluoromethyl)sulfonyl)amide (0.09 g, 2.76 × 10^−4^ mol, 1.1 equivalents) were dissolved in methanol in a round-bottom flask (50 mL). The reaction mixture was stirred at room temperature for 24 h. The solvent was evaporated under reduced pressure and the resulting crude residue was extracted with CH_2_Cl_2_ (3×), dried with anhydrous MgSO_4_, filtered, and concentrated, leading to compound **4** as a yellow oil (0.10 g, 62.45%). M.p. −36 °C. [α]${}_{D}^{25}=$ +23.15 (c = 0.034, MeOH). IR (ATR): ν_max_ = 3381, 3147, 3111, 2966, 2938, 2876, 1677, 1548, 1348, 1184, 1134, 1053 cm^−1^. ^1^H NMR (CDCl_3_) δ = 8.95 (s, 1H), 7.74 (s, 1H), 7.11 (s, 1H), 7.08 (s, 1H), 5.73–5.66 (m, 1H), 5.08 (d, *J* = 10.2 Hz, 1H), 4.97 (d, *J* = 17.1 Hz, 1H), 4.65 (d, *J* = 10.4 Hz, 1H), 4.30–4.08 (m, 2H), 3.27–3.19 (m, 1H), 3.16–3.03 (m, 1H), 2.68–2.44 (m, 2H), 2.39–2.21 (m, 1H), 1.46–1.39 (m, 2H), 1.33–1.20 (m, 2H), 1.00 (d, *J* = 6.6 Hz, 3H), 0.87–0.79 (m, 3H), 0.69 (d, *J* = 6.7 Hz, 3H).^13^C NMR (CDCl_3_): δ = 166.5, 135.5, 131.5, 123.6, 122.0, 121.5, 121.0, 120.3, 118.5, 115.9, 68.7, 49.5, 39.7, 34.3, 32.3, 30.8, 19.9, 18.6, 18.0, 13.5 ppm. MS (ESI+): 278.4 [M+H^+^, 100]. MS (ESI-): m/z (%) = 280.2 [NTf_2_^−^, 100]. C_18_H_28_F_6_N_4_O_5_S_2_: calcd. C 38.71, H 5.05, N 10.03, S 11.48; found C 39.07, H 5.31, N 9.70, S 11.03.

### 2.3. Column Preparation

Ten meters of capillary columns were coated using the static method at 40 °C, using solutions containing AAIL **4** (0.36%, *w*/*v*) and divinylbenzene (0%, 10%, 20%, 30%, and 40% of the amount of IL used) in dichloromethane. Prior to adding the solvent to the mixtures, 3 mg of AIBN [2,2′-azobis(2-methylpropionitrile)] was added. Capillaries were filled with the solution of the crosslinker, the monomeric ionic liquid, and the initiator. After coating, the ends of the capillaries were sealed, and the capillaries were introduced in a GC oven at 80 °C for 5 h in order to ensure complete polymerization. Helium gas was then flushed through the capillaries at a rate of 1 mL min^−1^ and the capillaries were conditioned overnight from 50 °C to 120 °C at 3 °C/min. Each column was prepared in triplicate and the efficiencies of the columns were tested with naphthalene at 100 °C. All columns showed efficiencies of 1700 plates/m or better.

The method used for the evaluation of AAIL columns was the Abraham solvation parameter model as this is the most comprehensive method available today [47,48].

This model is described by Equation (1).
*Log k* = *c* + *eE* + *sS* + *aA* + *bB* + *lL*(1)

Here, the higher-case letters *E*, *S*, *A*, *B,* and *L* are solute descriptors and are defined as *E* for the excess molar refraction, *S* for the solute dipolarity/polarizability index, *A* and *B* for the solute H-bond acidity and basicity, respectively, and *L* for the gas-hexadecane partition coefficient. The system constants (*e*, *s*, *a*, *b*, *l*) are used to describe the solvation properties of the stationary phase and are defined as follows: The *e* term represents π and nonbonding electron interactions, *s* the ability of the phase to interact with dipolar/polarizable solutes, *a* and *b* define the solvent hydrogen bond basicity and acidity, respectively, *l* includes dispersion forces (positive contribution) and a cavity term (negative contribution), and *c* is the system constant.

In order to evaluate the capillary columns, each individual probe molecule (dissolved in dichloromethane) was injected into the columns at three different temperatures: 60 °C, 90 °C, and 120 °C. The probe molecules shown in Table 1 were used for this purpose.

Multiple linear regression analysis and statistical calculations were performed using the Statgraphics Centurion XV version 15.2.06.

All characterization and separation experiments were performed using a Shimadzu GC-2010 Gas Chromatograph (Shimadzu, Kyoto, Japan) equipped with a flame ionization detector. Analysis of the Grob test was performed with helium as the carrier gas at a flow of 1 mL min^−1^ and a split ratio 100/1. The inlet temperature was maintained at 250 °C and the FID was held at 280 °C. The temperature was programmed as follows: 45 °C for 1 min, increased at a rate of 10 °C min^−1^ to 160 °C, and then held at this temperature for 2 min.

## 3. Results and Discussion

### 3.1. Optimization of Film Thickness Capillary Columns

A series of silica-supported polymeric ionic liquids-based stationary phases were prepared as capillary columns (see experimental section) using the polymerization of the *L*-valine-derived monomer AAIL **4** and DVB as the crosslinker for the coating of the silica surface (Figure 1). Column efficiency was evaluated using naphthalene as a soluble probe. Initially, several columns were prepared by varying the percentage of AAIL **4** in the absence of any crosslinking agent to form the corresponding linear PILs for the coating. As can be seen in Figure 2a, a percentage of 0.36% *w*/*v* AAIL provided the best separation efficiency. This percentage corresponds to a linear PIL film thickness of 0.23 microns. Therefore, capillary columns were prepared with 0.36% *w*/*v* of AAIL **4** for all subsequent studies.

In order to better understand the effects of the cross-linker on the efficiency of PIL-based stationary phases, PILs with varying amounts of DVB as the cross-linker (i.e., 0, 10, 20, 30, 40, and 60% *w*/*w*, Figure 1) were prepared and their efficiency compared. The efficiencies obtained are shown in Figure 2b.

As can be seen, all coated PIL columns possessed efficiencies of at least 1700 plates per meter. Moreover, the resulting columns were more efficient in the presence of the crosslinker and the efficiency increased with the amount of divinylbenzene added up to 20% *w*/*w* of DVB (DVB:AAIL ratio) with an efficiency of 2700 plates per meter at 20% DVB that compares well with efficiencies shown by other IL-based GC columns [35,49]. From this point, the number of plates per meter decreased significantly. This can be ascribed to the fact that for crosslinking degrees higher than 20% DVB, the interaction sites from AAIL located at the highly crosslinked regions become less accessible for the interaction with the analytes. Thus, some crosslinking is favorable to establish structurally well-defined and cooperative interaction sites, but polymerization with high crosslinking degrees can contribute not only to generate diffusional problems in the polymeric matrix but also to force less well-structured interaction sites, as has been demonstrated in polymer-supported catalysts [50,51].

### 3.2. Thermal Stability

Thermal stability determines the operation column temperature range and the lifetime of the GC columns. It is well known that the use of crosslinker agents can affect the thermal stability of PILs-based stationary phases [52]. Thus, the thermal stability of the prepared PILs-based columns was evaluated using two different methods. The first approach was a temperature-programmed GC method that can provide a rough idea of the thermal stability of the PILs-based columns studied [53]. In this case, 5-m × 0.25 mm fused silica capillaries were coated and subjected to increasing temperatures (from 50 to 400 °C at 3 °C min^−1^), and the column bleeding was measured until decomposition was observed. As can be seen in Figure 3, different behaviors were detected. For PILs containing very low crosslinking levels (PIL00 and PIL10), bleeding was observed at temperatures below 250 °C, which suggests some limitations for their application in GC. On the contrary, PILs with crosslinking degrees ≥ 30% displayed a similar behavior with very good thermal stability and no bleeding below 275 °C, As a matter of fact, the noncrosslinked PIL column (PIL00) showed the lowest thermal stability (bleeding starting at 225 °C) and, although not linearly, the incorporation of the crosslinker increased the thermal stability of the PIL-based stationary phases, reaching a maximum for the columns containing a higher degree of DVB (lack of bleeding up to 285 °C). It must be noted, however, that the efficiency decreases significantly, as mentioned above (see Figure 2b), for crosslinking degrees higher than 20% *w*/*w*. In this regard, the column based on the polymer PIL20, with 20% DVB, and displaying the higher efficiency, was thermally stable up to ca. 260 °C, indicating good thermal stability for standard GG applications. This method only gives a rough idea of the thermal stability of the columns studied. Thus, to properly evaluate the thermal stability of the columns, a second alternative set of experiments was carried out using an isothermal method.

Thus, the second approach was an isothermal method using naphthalene in the 100–350 °C temperature range. This method is based on holding the GC column at a series of constant temperatures for 12 h and measuring the retention of a well-retained probe compound at a lower temperature, before and after the isothermal conditioning. Table 2 shows the isothermal measurements of naphthalene at different temperatures with the six PIL-GC columns assayed. As can be seen, the retention factor for naphthalene was smaller for the column based on the noncrosslinked linear PIL (PIL00) increasing with the % of DVB for a given temperature. Furthermore, the results confirmed that an increase in the percentage of crosslinker led to an improvement in the thermal stability of the column, with an increase from 230 °C to 300 °C for columns with 10% to 60% of DVB, respectively.

### 3.3. Comparison of Interaction Parameters of PILs-Based Stationary Phases

The six different columns prepared were studied using the Abraham model to determine their solvation parameters, and to obtain information on the specific interactions that can be provided [47,48]. Although the resulting properties must be assigned to the two components of the composite, the fused silica, and the polymeric ionic liquid, changes in those properties must be ascribed to changes in the PIL and, more specifically, to the modification in the crosslinking degree, which represents the most relevant structural change. It must be noted that the stationary phase containing 60% of DVB (PIL60) could not be properly characterized, due to the lack of elution of most of the compounds at 60 °C and 90 °C and for the excessive asymmetry of the peaks observed, which generated large errors in the Abraham’s model. The values of all the solvation parameters calculated for the other stationary phases are shown in Table 3. As can be seen, a slight decrease in the system constants was observed when increasing the temperature for all stationary phases, as usually occurs in gas chromatography analysis [47,48,52,54]. In all cases, the hydrogen bond basicity (a-term) and the dipole interactions (s-term) are the dominant system constants, followed by the hydrogen bond acidity (b-term) and the term involving dispersion forces (l). In imidazolium ILs, the structure of the cation and anion involved determines their acid/basic properties, as well as the values obtained for other parameters [54]. As the nature of both the cation and the anion in the PIL component remain constant for all the columns studied, the differences observed must be assigned to the crosslinking of the polymers, most likely associated with changes in the mobility and accessibility of the functional groups in the PIL coating, and to the possible generation, for the higher crosslinking degrees, of polymer domains with different polarities, which could also contribute to the low performance of PIL60.

The hydrogen bond basicity (a-term) in imidazolium ILs has been shown to be essentially dependent on the nature of the anion. The value of this parameter for the five different columns is relatively similar as they share the NTf_2_ anion for the imidazolium cation (ca. 1.77 ± 0.13 at 60 °C) and is also similar to the value calculated for ILs containing this anion (i.e., 1.752 for BMIM-NTf_2_ at 70 °C) [54]. A value of 1.377 has been reported for a commercial IL-based column (1,9-di(3-vinylimidazolium) nonane bis(trifluoromethyl sulfonyl)imide at 120 °C [23], a value close to that found in our case (1.23 ± 0.22), The hydrogen bond acidity (b-term), on the other hand, is different from zero for all stationary phases studied. This is relevant in comparison to most commercial stationary phases (i.e., HP-5 and ZB-WAX), where the b-term is practically zero [47,48,55], and represents a clear competitive advantage of these columns containing composite materials based on polymeric ionic liquids from AAILs. The presence of acidic hydrogen atoms at positions 2, 4, and 5 of the imidazolium ring contributes to the acidity observed in this family of ILs. In the present case, however, the value observed is significantly higher (0.66–0.90 at 60 °C) than that reported for BMIM-NTf_2_ (0.378 at 70 °C) [54]. The presence of AAIL **4** of an amide functionality providing an additional hydrogen bond donor group and preserved in all prepared PILs can explain this fact. A similar value for the b-term has been reported for an imidazolium bistriflamide salt displaying a hydroxyl group [55], and for the commercial 1,9-di(3-vinylimidazolium) nonane bis(trifluoromethyl sulfonyl)imide-based column displaying ditopic bisimidazolium units [23]. It is interesting to note that the value for this parameter (b-term) initially decreased with the crosslinking (for 10 and 20% DVB), which can be attributed to the limitation in mobility and accessibility of the acidic sites in the crosslinked polymeric matrix. However, for higher degrees of crosslinking (30 and 40%), the value of the b-term increased significantly at low temperatures (up to 0.90 at 60 °C for 40% DVB). The partial generation of polymer domains with different polarities upon increasing the DVB content at those levels can be responsible for this phenomenon. At higher temperatures, the b-term always decreased with the crosslinking, as the increased chains mobility will reduce domain separation. In all cases, the value of this parameter decreased with temperature and, at 120 °C, fully inversely correlated with the DVB content, as could be expected.

The e-term shows the ability of the PIL-silica composites to interact with solutes through π- and nonbonding electron interactions, being positive when the stationary phase interacts with the solutes through the electron lone pairs of the present heteroatoms and/or the π-cloud of the aromatic rings present. For all the columns studied, the e-term was very small or slightly negative, indicating that there is no appreciable interaction between the probe solutes and the stationary phases through π- or n-electrons. A value of zero has been calculated for this term in the case of BMIM derivatives not containing electron-donating substituents at the imidazolium ring nor strongly coordinating anions [54]. In the present case, this indicates that the aromatic rings of the polymeric matrix (from DVB units) do not significantly contribute to this kind of interaction. PIL20 containing 20% of DVB and showing slightly positive values at 60 °C is the stationary phase displaying the lowest variation in this parameter with temperature, suggesting again a high degree of homogeneity in the polymers formed. In the case of the l-term, the stationary phases studied had l > 0 coefficients, implying that positive dispersion interactions dominate. As can be seen from the higher values obtained for the crosslinked stationary phases, the structural components originating by the polymerization of DVB provide a clear component for this term.

### 3.4. Chromatographic Performance of the Mono and Cross-Linked Ionic Liquid Stationary Phase

In order to obtain additional information on the morphologies and physical-chemical properties of the PIL-based columns prepared, as well as on their potential for efficient chromatographic separations, the Grob test was injected into each of the prepared stationary phases and the results obtained were analyzed. The Grob test contains a mixture of compounds of different nature, and is used to evaluate the separation efficiencies, the acid and basic characteristics, and the relative polarity of chromatographic columns [56,57]. The mix contains various types of organic compounds including hydrocarbons, fatty acids methyl esters, acids, bases, and alcohols, and each peak gives specific information on the column. Alkanes (n-decane and n-undecane) represent a 100% recovery marker and should have symmetrical peaks if the column was correctly prepared. Alcohols (1-octanol and 2,3-butanediol) are used to measure the adsorption caused by hydrogen bonding mechanisms. The acid/base interactions are shown by the behavior of 2,6-dimethylaniline, 2,6-dimethylphenol, dicyclohexylamine, and 2-ethylhexanoic acid. Fatty acid methyl esters (methyl decanoate, methyl undecanoate, and methyl dodecanoate) are a homologous series of fatty acid methyl esters used to determine the separation efficiency of the column. Thus, the use of this mixture of compounds of different nature can allow the determination of the influence of the DVB content in the polymerization mixtures on the general properties of the column and also to assess their suitability as GC columns for analytes of different nature.

The chromatograms obtained with columns PIL40 and PIL60, based on mixtures of the vinylic AAIL and 40% and 60% of DVB, respectively, are not shown and will not be discussed as an appropriate separation of the compounds was not observed. Figure 4 shows the results obtained for the other PILs. For a proper comparison, the chromatograms have been represented at the same scale. Figure 4a,b display the results for the stationary phases prepared in the absence of crosslinking (PIL00) and with just 10% of DVB (PIL10), respectively, and reveal that all compounds exhibited high peak tailing. This indicates that the coating films generated throughout the silica capillary are not homogeneous, and that nonspecific adsorptions with residual accessible silanols can be present. Furthermore, 1-nonanal and 1-octanol show both peak tailing and reduced peak heights, which implies that these stationary phases are highly polar with H-bond accepting capabilities.

Figure 4c, representing the results for the stationary phase prepared with 20% DVB (PIL20), shows the best peak symmetries and less mass transfer losses, which is in agreement with the previously discussed solvation parameter coefficients results and with the results for the calculation of the theoretical plates number. In this column, the alkanes were better-separated from the solvent peak as compared with the other columns studied, and the fatty acid methyl esters were separated with good symmetries and good separation efficiencies. Moreover, better peak symmetries and lower mass transfer losses were also observed for 2,3-butanediol, 2-ethylhexanoic acid, and dicyclohexylamine using the PIL20 column than using the commercial imidazolium IL82 and IL100 columns [23]. This suggests that this column can be of practical application to quantify low levels of compounds with a high capability to interact through H-bonding. It must be noted that the results were again worse when a higher crosslinking degree was used (PIL30, 30% DVB, Figure 4d), the corresponding chromatogram displaying lower separations and less symmetric peaks.

Although not all the analytes were separated using these PIL columns, it should be noted that in all columns studied, 2,3-butanediol and 1-octanol eluted after decane and undecane. This indicates that the interactions through hydrogen bonds between such PILs-DVB stationary phases and the analytes play an important role in the chromatographic separation. Another notable fact is that 2,6-dimethylphenol always eluted after 2,6-dimethylaniline, indicating that these stationary phases retain acidic analytes more strongly than basic analytes, as suggested by the values of the corresponding Abraham parameters. Similar behavior was observed with the commercial imidazolium IL82 and IL100 columns, although 1-octanol eluted before 2,3-butandiol using these commercial stationary phases [23]. Finally, it should be emphasized that 2-ethylhexanoic acid always eluted before FAMES, opposite to commercial IL82 and IL100 stationary phases.

## 4. Conclusions

In this study, five stationary phases containing polymeric ionic liquids based on the AAIL **4** and variable amounts of DVB as the crosslinker were prepared, and their thermal stabilities and chromatographic properties for GC analysis were evaluated. The results obtained demonstrate that DVB is an excellent modulator of the properties of the resulting polymeric phases, and the degree of crosslinking represents a key parameter to define the overall chromatographic efficiency of the resulting silica-coated columns. In this study, the best results were obtained for a 20% crosslinking (DVB content, PIL20). The thermal stabilities observed for the cross-linked stationary phases were always higher than those for the stationary phase just containing the linear AAIL **4**-based PIL (PIL00). Moreover, the introduction of a crosslinker such as DVB was able to improve the separation properties, most likely due to the formation of a more homogeneous film along the capillary. However, the amount of crosslinker needs to be carefully adjusted. DVB amounts over 20% led to a loss of the dual behavior of the stationary phases most probably due to a reduction in the homogeneity of the polymeric phase with generation of separate domains (apolar/polar) and to a reduction in the accessibility to the analytes of the AAIL subunits through a reduction in mobility and diffusional properties.

The solvation parameters calculated show that the dominant interactions are the hydrogen bond basicity (a-term) and the dipole interactions (s-term), with some contributions from the hydrogen bond acidity (b-term) and the dispersion forces (l-term), a trend that has also been observed for other imidazolium ILs. It must be noted that the l-term for the stationary phases based on the crosslinked PILs was greater than those for the linear polymeric coating exclusively based on the AAIL monomer (PIL00). This constant decreased with the increase in the amount of DVB. The highest values of the b term as compared with other commercial columns provides a significant potential for developing stationary phases with selectivities different from those currently available.

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
