# Peer review of "Polymeric Ionic Liquids Derived from L-Valine for the Preparation of Highly Selective Silica-Supported Stationary Phases in Gas Chromatography"

_polymers, 2020, doi:10.3390/polym12102348_

Round 1

Reviewer 1 Report

Purity of the used compounds such as for example divinylbenzene and dichloromethane should be reported.

Other conditions of the realized experiments are described correctly.  The methods are adequately described.The staistical analysis was done in the appropriate way. The results and discussion are clearly presented.

Reviewer 2 Report

The authors described an interesting poly(ionic liquid) platform for gas chromatography. The manuscript is well-organized and written. I recommend its publication after the following minor modification.

1 For the application of ionic liquid in the introduction part, the related literature about the application of ionic liquid in materials could be cited, ACS Applied Materials & Interfaces,2017, 9(8), 7217-7223.

2 What is FID?

3 For the synthesis, why did the authors choose 40 centigrade degree as the used temperature?